# The Role of Intrinsically Disordered Proteins in Liquid–Liquid Phase Separation during Calcium Carbonate Biomineralization

**DOI:** 10.3390/biom12091266

**Published:** 2022-09-09

**Authors:** Aneta Tarczewska, Klaudia Bielak, Anna Zoglowek, Katarzyna Sołtys, Piotr Dobryszycki, Andrzej Ożyhar, Mirosława Różycka

**Affiliations:** Department of Biochemistry, Molecular Biology and Biotechnology, Faculty of Chemistry, Wroclaw University of Science and Technology, Wybrzeże Wyspiańskiego 27, 50-370 Wroclaw, Poland

**Keywords:** liquid–liquid phase separation, intrinsically disordered proteins, biomineralization, calcium carbonate, otoliths, nucleation pathways

## Abstract

Some animal organs contain mineralized tissues. These so-called hard tissues are mostly deposits of calcium salts, usually in the form of calcium phosphate or calcium carbonate. Examples of this include fish otoliths and mammalian otoconia, which are found in the inner ear, and they are an essential part of the sensory system that maintains body balance. The composition of ear stones is quite well known, but the role of individual components in the nucleation and growth of these biominerals is enigmatic. It is sure that intrinsically disordered proteins (IDPs) play an important role in this aspect. They have an impact on the shape and size of otoliths. It seems probable that IDPs, with their inherent ability to phase separate, also play a role in nucleation processes. This review discusses the major theories on the mechanisms of biomineral nucleation with a focus on the importance of protein-driven liquid–liquid phase separation (LLPS). It also presents the current understanding of the role of IDPs in the formation of calcium carbonate biominerals and predicts their potential ability to drive LLPS.

## 1. Introduction

Liquid–liquid phase separation (LLPS) is one of the key mechanisms affecting how macromolecular assemblies, including membrane-less organelles (MLOs), are formed and regulated [1]. This reversible, thermodynamically-driven process relies on the separation of a homogeneous solution into two distinct liquid phases with different concentrations of solutes. LLPS occurs as a two-phase system based on the concentration of molecules and the physio-chemical parameters of the microenvironment [2]. Phase-separated condensates, especially MLOs, are multicomponent assemblies of proteins and other macromolecules, e.g., nucleic acids [2,3,4]. The interactions between the components are weak, transient, and multivalent [5,6]. The proteins that are found to reside in a condensate may play diverse roles (e.g., scaffold, co-scaffold, clients, and regulators) in maintaining the condensate integrity, composition, and biochemical properties. Scaffolds can self-associate and drive LLPS, so they are primarily responsible for condensate formation. Clients, on the other hand, are low-valency molecules that are recruited to the condensate through their interactions with scaffold biomolecules [7]. Their content may be adjusted to the changing conditions within and outside the condensate [8].

LLPS as a physical process has been known for decades in polymer science, but it has been rediscovered in eukaryotic cells [9]. At present, it appears that it is a universal phenomenon that plays an important role in the interior organization of eukaryotic cells, in the formation of MLOs in prokaryotes [10,11], and during viral life cycles [12]. Notably, some extracellular protein interactions facilitate LLPS [13]. Biomineralization is the process by which organisms produce minerals under biological control. The control of biomineralization is aimed at creating specific minerals composed of inorganic and organic fractions. A very common inorganic component of biominerals is calcium carbonate in the form of various, usually non-calcite polymorphs [14,15]. Although the mechanisms of biomineral nucleation processes have been studied for years, their principles are still enigmatic. Calcite, the most stable polymorph of calcium carbonate, serves as a model for the primal and accepted-for-years theory of crystal growth, known as a classical theory. Some recent studies, however, showed that the formation of calcium carbonate frequently does not follow the classical model [16,17]. Since 2000, when Gower et al. launched the polymer-induced liquid precursor (PILP) concept of biomineral precursors, it has been widely accepted that the early events of biomineral formation may follow diverse alternative pathways [18]. Moreover, further experimental results concerning calcium carbonate mineralization presented prenucleation clusters as a key precursor phase in mineral formation [19]. Currently, a concept that involves the formation of dense liquid precursors of amorphous calcium carbonate (ACC) via LLPS has become a popular topic of investigation [20,21].

As indicated above, the molecular and biochemical mechanisms involved in the biomineralization pathway remain puzzling. Additionally, the significance of intrinsically disordered proteins (IDPs), which are an abundant organic component of hard tissues [22], in the formation of liquid precursors of biominerals remains to be solved. Research on the interactions between proteins and divalent cations is essential for understanding the resulting liquid precursors. In the available literature, there are only a few examples describing such interactions. However, they may help to understand the functional and pathological phase behaviours in the biomineralization process.

In this paper, the role of proteins, mainly IDPs, in the formation of calcium carbonate biominerals is reviewed. We focus on different mechanisms of mineral formation and discuss the potential role of LLPS in the nucleation processes. Since the discovery of protein-driven phase separation [9,23], the concept of LLPS has been deeply integrated into the life sciences, especially into research focusing on cell biology. Phase separation has also been considered in studies concentrating on mineralization processes, yet often the term, phase separation (and LLPS in particular), is masked by coalescence, whereas liquid condensates/droplets are often referred to as coacervates. In this review, we gathered the knowledge that is scattered and hidden under non-uniform terms. We also discuss the potential importance of IDPs in LLPS during calcium carbonate biomineralization.

## 2. Role of Proteins and Divalent Ions in LLPS

First, it was suspected that strong stereospecific interactions between protein components played a major role in the formation of MLOs. However, subsequent studies have shed light on the importance of very weak interactions, e.g., electrostatic, hydrophobic, and π-π interactions [5]. Currently, we know that both strong and weak interactions, which occur simultaneously, contribute to the entire complex interaction network and facilitate condensate formation. Additionally, the interaction of proteins with solvent plays a critical role in the regulation of phase transitions; thus, an important feature affecting protein-induced LLPS is solubility [24]. Most proteins that undergo LLPS have poor solubility in water. Placing such proteins in the structure forming during phase separation is more energetically beneficial than allowing them to come into contact with water [25]. This is particularly important in the case of proteins containing low sequence complexity and a richness in residues that tend to aggregate [26]. Since the physio-chemical properties of the solvent strongly impact protein solubility, LLPS occurs as a function of parameters such as osmolality, ionic strength, pH, or temperature [2,27,28,29].

Another key factor underlying LLPS is multivalence, i.e., the availability of many different binding sites in the molecule. Multivalent proteins can form heterologous electrostatic interactions with different, oppositely-charged proteins or homologous interactions between their repetitive domains [30]. Multivalent proteins have a critical phase separation threshold that is often related to the number of domains it contains and the availability of ligands [25]. Multivalence is especially characteristic for IDPs; therefore, this class of proteins is often involved in promoting phase separation. IDPs do not fold into unique, three-dimensional globular structures under physiological conditions. Changes in the cellular environment and conformational properties allow IDPs to take on numerous conformations induced by the attachment of ligands, binding to the membrane surface, or various types of post-translational modifications [31]. NMR analyses of IDPs after LLPS show that disordered regions of proteins retain conformational flexibility in the condensed phase [27]. Interestingly, IDPs can also form complexes with other macromolecules or metal ions and consequently undergo, at least in fragments, disordered-to-ordered transitions [32]. IDPs with an inherent propensity for LLPS affect various cellular functions, such as signaling, cell division, intracellular transport, cell cycle control, and regulation of transcription and translation. Unfortunately, in some cases, structural features of IDPs can promote the formation of abnormal conformations prone to aggregation, which in turn causes severe diseases associated with protein misfolding, such as Alzheimer’s, Parkinson’s, or Huntington’s disease [33]. Interestingly, not all fibrous structures cause disease. Amyloid aggregation of a large number of IDPs is associated with the biogenesis of functional amyloids, which positively influence various biological functions, e.g., melanin pigment formation, bacterial biofilm formation, or biominerals [34].

Recently, it was shown that divalent cations also have the ability to tune protein phase behaviour. However, it remains a largely unexplored area. The first report describing LLPS of proteins in the presence of divalent cations comes from 2020. Singh et al. showed that LLPS of tau protein is modulated by zinc ions, which strongly enhance the propensity for tau to undergo LLPS by lowering the critical concentration of protein [35]. Surprisingly, none of the other divalent metal ions tested (manganese(II), iron(II), cobalt(II), nickel(II), and copper(II)) were found to promote the phase separation of tau. However, the mechanism by which zinc ions promote LLPS of tau is not known. Singh et al. suggested that local folding of tau, resulting from zinc binding, could cause an increased density of positive and negative charges within particular regions. This, in turn, would lead to stronger attractive intermolecular interactions, facilitating LLPS [35]. Another proposed theory is that zinc ions promote LLPS of tau by facilitating the formation of transient intermolecular cross-links between protein molecules [35]. However, these suggestions need to be further studied.

Divalent cations can modulate phase transitions both directly and indirectly through interactions with other proteins. EF-hand domain protein 2 (EFhd2) is a conserved calcium-binding protein. It is expressed in various tissues but predominantly in the central nervous system [36,37]. EFhd2 has been found to be associated with tau aggregates in the mouse model, JNPL3, and as a tau-associated protein in Alzheimer’s diseased brains [36,38]. Recent studies have shown that EFhd2 modulates the phase transition of tau and directly alters tau liquid phase behaviour to form solid-like structures in vitro, and this phenomenon is controlled by calcium ions [39]. Notably, both EFhd2 and tau, in the absence of calcium ions, lead to the formation of solid-like structures containing both. On the other hand, in the presence of calcium ions, EFhd2 and tau phase separate together into liquid droplets [39].

Divalent cations can also modulate the LLPS of transcription factors. It was shown that zinc and copper(II) ions induce LLPS of the F region of the *Aedes aegypti* ecdysteroid receptor [40]. Since this region seems to affect the dimerization of nuclear receptors, the interactions with other proteins, and the stabilization of ligand binding, LLPS of the ecdysteroid receptor might contribute to the regulation of transcriptional activation.

Protein interactions driving LLPS may vary depending on the nature of the amino acid sequence [41]. Proteins are polyelectrolytes that can have both positive and negative charges [42]. Well-described examples of LLPS (often referred to as coacervation) are those based on interactions between polycationic proteins and polyanionic RNA molecules [43]. Less is known concerning the ability of polyanionic proteins to undergo LLPS in a similar charge-dependent manner. Mayfield et al. identified a previously unknown mechanism of calcium-dependent LLPS occurring within the endoplasmic/sarcoplasmic reticulum (ER/SR) that explains efficient calcium ion buffering and storage [44]. It was shown that calcium ions modulate LLPS of the polyanionic protein and major calcium-binding protein of the SR of skeletal muscle, calsequestrin-1 (CASQ1). The process was reversible and occurred within cells. CASQ1 is an IDP that influences its capacity for LLPS. It was also shown that the LLPS of CASQ1 is regulated via phosphorylation by the secretory pathway kinase Fam20C, which phosphorylates structurally-conserved regions of CASQ1 [44]. Thus, the phosphorylated protein (pCASQ1) more readily entered the LLPS state. Mayfield et al. [44] suggested that this likely arises from the increased disorder and conformational flexibility of pCASQ1. Additionally, they hypothesized that calcium-dependent LLPS of polyanionic IDPs is a widespread and evolutionarily-conserved phenomenon that might represent a major mechanism underlying calcium ion handling and signaling [44].

## 3. Intrinsically Disordered Proteins in Biomineralization

Although biominerals are built on a scaffold of collagenous proteins, IDPs are the most prominent regulatory effector of the process. IDPs, with no defined secondary structure, act as specific regulators of biomineral formation by influencing their nucleation and orienting their growth and controlling polymorph selection (Figure 1). It is estimated that proteins involved in biomineralization processes in humans and other organisms (e.g., fish, mollusks, and diatoms), as well as in the formation of eggshells, have a high average disorder level of 53% [22,45].

The basis for the biomineralization activity of IDPs lies in their ability to interact with their inorganic mineral counterparts. Proteins involved in biomineralization can be incorporated into biominerals at the inter- and intracrystalline levels [46]. The protein–mineral interphase, in the case of IDPs, is mainly governed by electrostatic interactions resulting from the characteristic composition of the protein primary sequence and post-translational modifications [47]. IDPs show an increased frequency of acidic amino acids, such as Asp, Glu, and commonly phosphorylated Ser, compared to the average in the SwissProt database [48]. The first stages of calcium carbonate biomineral nuclei formation involve the uptake of calcium ions from the environment by negatively-charged amino acid side residues. In this way, IDPs form a specific link between the organic matrix and the inorganic part of the biominerals. This linkage is further enhanced by the frequent phosphorylation of biomineralization-related IDPs [47].

The composite character of all biominerals is their undeniable advantage. The small presence (up to 5% *w/w*) of organic matrices often containing hundreds of types of proteins affects their mechanical properties [49,50]. In nacre, the improvement of mechanical properties, such as high resilience, ductility, and energy dissipation, is attributed to the presence of a thin layer of organic components that counterbalances crack propagation [48,51]. An in vitro study on the effect of disordered Otolith Matrix Macromolecule 64 (OMM-64) on the biomineralization of calcium carbonate also confirmed that IDPs enhanced the mechanical properties of crystals. Compared to non-bionic calcium carbonate, biominerals with embedded OMM-64 protein showed improved properties, such as greater flexibility, as determined by atomic force microscopy (AFM) [52]. The ability to modulate crystal growth, morphology, or mechanical properties is associated with intra- and intercrystallite interactions at the IDP-mineral level. These interactions are mediated by electrostatic interactions between inorganic ions and side residues of acidic amino groups or functional groups [47]. The increased participation of acidic amino acids in the IDP sequence is likely to mediate protein incorporation into the crystal structure, as in the case of Asp, which was shown to replace bicarbonate ions with its carboxyl group in biominerals [53].

In addition to the characteristic features of the amino acid residue chain, biomineralizing IDPs also exhibits a number of post-translational modifications, including extensive phosphorylation and glycosylation. One well-studied glycosylated IDP involved in the sea urchin biomineralization process is spicule matrix protein (SM30) and its various isoforms [54,55]. Extensive phosphorylation on the other hand was confirmed for zebrafish Starmaker (Stm) protein in a proteomic study, as residues 25 Ser and 3 Thr were identified as phosphorylated. Notably, this protein was the only one in this study to have this type of modification [50].

The attachment of phosphate groups to biomineralization-related IDPs influences their molecular properties, changing their isoelectric point and making them even more acidic [56]. This modification increases the ability to bind calcium ions more effectively. The comparative studies on Stm protein mineralization activity with and without phosphorylation indicates that phosphorylated Stm reduces the size and influences the morphology of the crystals that are greater than the non-modified version of Stm [57].

In in vitro studies, casein kinase 2 is typically used for phosphorylation [57], nonetheless the proteomic studies identified Fam20C kinase as an important biomineralization player [49,58]. Fam20C was confirmed to phosphorylate small integrin-binding ligand and N-linked glycoproteins (SIBLINGs) in bone, and the mutations of Fam20C result in bone dysfunction [58]. The same kinase was identified in fish otoliths [49] and the shells of *Pinctada fucata* oysters, where its expression is enhanced in the shell repair process [59].

The modifications of biomineralization-related proteins, both phosphorylation and glycosylation, alter the electrostatic interactions with inorganic counterparts of the biominerals. However, to define in one sentence the role of glycans in biomineralization-related proteins is beyond the means of research and nature, given the possible variations and heterogeneity of this modification [55]. In the case of recombinant aragonite protein 24 (AP24) nacre protein, the glycosylated AP24 influences the directions of crystal growth and inhibits the nucleation process, while non-glycosylated proteins stabilize the mineral phase more efficiently. At the same time, the lack of modification does not influence the hydrogels formation activity [60]. In the case of SpSM30B/C, the influence on biomineralization remains similar in both glycosylated and non-glycosylated variants; nonetheless, the non-glycosylated SpSM30B/C does not form aggregates effectively, contrary to the glycosylated version [61].

## 4. The Role of Proteins in Formation of Calcium Carbonate

To illustrate the importance of individual IDPs in the biomineralization of calcium carbonate, it is worth reviewing the functions of specific proteins among different taxa. One of the most widely-understood proteins from molluscan shells is aspein. This protein consists of four regions, including a signal peptide (cleaved before the protein is released to the extracellular matrix), an uncharged N-terminal region, and two domains enriched in Asp, i.e., a DA and poly-D domain. The Asp-rich domains are crucial for the regulation of biomineral formation [62]. Aspein promotes calcite formation over aragonite. It was shown that the protein binds to calcium ions with higher affinity than it binds to magnesium ions, thus increasing the local concentration of calcium ions and inducing calcite formation [62,63]. Additionally, the Asp-rich domains are not conserved between species, suggesting that sequence specificity is not a determinant of aspein’s biomineralization activity [64].

In crustaceans, such as crayfish, one of the widely described-proteins is acidic calcification-associated peptide-1 (CAP-1), which is involved in the mineralization of the exoskeleton. CAP-1 has a dual structure: the N-terminal part is most likely responsible for maintaining the conformation of the protein, while the C-terminal part is associated with calcium carbonate mineralization [65]. The C-terminal part possesses calcium ion binding ability, which in turn also increases their local concentration. Moreover, the protein exhibits carbonate growth inhibitory properties, which are further enhanced by the presence of phosphoserine in the C-terminal part [66].

In stony corals, coral acid-rich proteins (CARPs) are a group of proteins involved in calcium carbonate precipitation. To date, the characterized proteins of this group are highly acidic and possess calcium-binding domains [61]. Their N-terminal region, which is extremely rich in acidic amino acid residues, is directly involved in calcium carbonate formation. Recombinant N-terminal fragments of CARP-1 and -2, as well as full-length CARP-3 and -4, are able to spontaneously precipitate calcium carbonate in an experiment with artificial seawater [67]. In addition, the ability to control crystal polymorphs was confirmed for the CARP-3 protein [68].

The aforementioned SM30 is a sea urchin spicule protein that exists in multiple isoforms. It is an acidic, glycosylated protein with a C-type lectin domain at the N-terminus and a disordered C-terminal region [69,70]. Studies on the B/C isoform hybrid of the SM30 protein from *Strongylocentrotus purpuratus* show that SM30 is capable of aggregating and forming a hydrogel that controls the biomineralization process by initially stabilizing ACC, thereby forming single crystals of calcite and promoting directed crystal growth [61].

Nonetheless, in our laboratory, we have extensively studied proteins related to the biomineralization of human otoconia and fish otoliths. These calcium carbonate biominerals located in semicircular canals of the inner ear are responsible for the sense of gravity, balance, the perception of linear and angular acceleration, pressure changes, and sound vibrations [71]. Otoconia, as in *Homo sapiens*, are similar to sand suspended in the gel mass of utricles and saccules, while fish otoliths resemble stones placed in three otolithic end organs: saccules, lagenas, and utricles. Otoconia remain inert during their lifespan, while otoliths grow diurnally, accumulating new layers of calcium carbonate and organic matrix [72,73,74]. The mineralization process takes place in an acellular medium, the endolymph, which is a fluid rich in structural materials such as calcium, proteins, and other macromolecules [75]. The organic matrix of human otoconia and zebrafish otoliths constitutes up to 5% of the biomineral mass [76,77]. Although the percentage content of the organic matrix is quite low, proteins, such as OMM-64, Stm, Starmaker-like (Stm-l), otolin 1, and otolith matrix protein-1 (OMP-1; the orthologue of mammalian otoconin 90 (Oc90)-Otoc1), are required for normal otolith growth [78,79,80,81,82].

Stm from zebrafish was the first protein found to be capable of controlling the process of calcium carbonate biomineralization. Stm controls the size, shape, and polymorph of the mineral component of the otolith [83]. Stm acts as an inhibitor of crystal growth; the decrease in crystal size depends on the protein concentration. The ability of Stm to act as a crystal nucleation factor and inhibitor of crystal growth is directly related to its degree of phosphorylation, which adds a negative charge and increases the binding affinity to calcium ions [57].

Stm-l, an IDP from medaka (*Oryzias latipes*), is able to adopt a more ordered and rigid structure under the influence of the environment and has a negative effect on the size of precipitating crystals. However, the higher number of crystals formed in the presence of the protein suggested that Stm-l could also act as a crystal nucleator [81]. According to Różycka et al., in vaterite crystals, the occurrence of Stm-l is probably limited to its nucleation site, whereas in calcite, the distribution of the protein occurs throughout almost the entire crystal. The time-dependent mineralization tests allow visualization of the sequential deposition of Stm-l in forming calcite. The protein acted as a nucleator of crystal growth through the condensation and formation of intermediate phases at the early stages of the process. Then, Stm-l regulated crystal growth by adhering to step edges on calcite, which resulted in ellipsoidal to spherical shapes of crystals and a reduction in crystal/crystallite sizes [84].

Similar to Stm-l, OMM-64, a highly-acidic IDP rich in Asp and Glu residues, can undergo transitions to more ordered states [80]. Additionally, the presence of calcium ions resulted in protein compaction. In vitro biomineralization experiments showed that OMM-64 plays a biological role similar to that of Stm and Stm-l and controls both the size and the shape of calcium carbonate crystals. As shown with two-photon fluorescence experiments, the enhanced density of the protein in the central part of the crystals suggested the participation of OMM-64 in the nucleation of calcium carbonate crystals. The nucleation of crystals can be initiated by the adsorption of calcium ions exposed to OMM-64 acidic residues and their local concentration, accompanied by the collapse of the protein molecule. Hyperphosphorylation of OMM-64 strengthens the inhibitory effect of the protein in the biomineralization process [52,80].

One of the major components of the otolith matrix is otolin-1, a 48 kDa collagen-like protein [78,85,86,87,88]. This secreted otoconin is present both in the organic matrix of human otoconia and zebrafish otoliths. In zebrafish, otolin-1 can be found in the otolith itself and on the boundaries of the structure as a link between the otolith and the sensory epithelia [85]. Otolin-1 is composed of four domains: a 23-amino acid signal peptide, a non-collagenous N-terminal domain, a central collagen-like domain, and a globular C-terminal C1q domain liable for protein molecule trimerization [89]. The presence of calcium ions influences the secondary and tertiary structure of recombinant otolin-1, especially the thermal stability [90]. Recombinant human and zebrafish otolin-1 forms high-order oligomers [90]. The oligomerization of zebrafish protein is dependent on its concentration and the presence of calcium ions, whereas human protein exhibits the same oligomeric stage regardless of these factors. Despite the high sequence similarity (45.51% identity and 56.58% similarity), these two homologues show differences that may be reflected in the nature of otoliths and otoconia [90]. Otolin-1 could be the crucial element of the organic matrix of otoconia and otoliths, serving as a high-order oligomeric scaffolding protein stabilized by calcium ions [90].

The process of otoconia and otolith growth involves a series of temporally- and spatially-specific events that are tightly coordinated by numerous proteins [73]. Otolin-1 and OMM-64 extracted from rainbow trout (*Oncorhynchus mykiss*) otoliths led to the formation of aragonite crystals in an in vitro biomineralization assay, while otolin-1 and OMM-64 separately induced small calcite and vaterite crystals, respectively [79]. Similarly, recombinant murine otolin-1 influenced the size and shape of the obtained crystals, but the effect was enhanced by Oc90 [91]. Interestingly, it has been shown that another IDP, dentin matrix protein 1 (DMP1), which is an extracellular matrix protein essential for the biomineralization of calcium phosphate in bone and dentin [92], was present in the inner ear, specifically in otoconia [93]. Later studies indicated that the 57K fragment of DMP1 [94] formed oligomers in the presence of calcium ions and affected the morphology of calcium carbonate crystals in vitro. These studies suggest that DMP1 shows a previously unknown regulatory function for the biomineralization of otoconia [95].

## 5. Calcium Carbonate Nucleation Process

The key features of biominerals, such as lattice orientation, particle size, and size distribution, are determined by the conditions prevailing during the first phase of crystal growth—nucleation. To this day, the process of nucleation of crystals from a solution remains poorly characterized mainly due to the difficulty of measurements and their interpretation at the low (atomic) level of matter organization, but also because in nature, nucleation involves a number of unknown or hardly characterizable factors. These factors include surface wettability or inhomogeneity, which affect the nucleation barrier and the nucleation rate. Another difficulty stems from the interdisciplinary nature of this phenomenon, which has caused diversity in the terminology used by researchers from different fields [20,96,97].

In general, the mechanisms aiming to explain the calcium carbonate nucleation kinetics from solutions can be divided into two distinct groups, i.e., classical nucleation theory (CNT) and alternative multistep (non-classical) pathways (Figure 2). The classical nucleation mechanism first introduced in 1878 by Gibbs [98] and developed during the past century [99,100,101] provides a fairly simple explanation of how crystals nucleate in homogeneous and heterogeneous pathways. This process is limited by the energy barrier resulting from the cost of generating a phase interface and, with it, the interfacial tension between the nucleus (also known as a cluster) and its surroundings. In homogenous nucleation, the process of nuclei formation is driven by the stochastic fluctuation of monomer association in the supersaturated solution, while in heterogeneous nucleation (most common in nature), the process is accelerated due to the presence of foreign molecules (including proteins), which can act as heterogeneous nuclei and reduce the free energy barrier [102]. Briefly, CNT assumes the presence of an unstable pre-critical cluster that grows by successive and reversible attachment (and detachment) of monomers to its surface that are components of the final crystal. As the cluster grows, the Gibbs free energy of the system increases, but only to the maximum value for the critical size of the nucleus and the formation of the metastable cluster. After exceeding the critical size, a stable solid form of the post-critical nucleus is formed, and the free energy is released during crystal growth. Smaller nuclei are thermodynamically unstable and dissolve again [19,96,103].

Recently, the development of advanced experimental and bioinformatics analyses has provided evidence that the formation of calcium carbonate frequently does not follow the CNT [16,17,104]. The alternative mechanisms are based on the observation of the formation of stable or metastable precursors, most likely created by the collisions and coalescence of their constituent components. These results clearly conflict with the nucleation picture presented in the CNT (Figure 2A). The presence of such individuals indicates the appearance of additional minima in the graph illustrating the Gibbs free energy of the calcium carbonate precipitation reaction (Figure 2B). This means that multistep nucleation pathways comprise more than one barrier that must be overcome, along with a number of local minima corresponding to the formation of precursors with different sizes and probably different structural arrangements [19,103,105]. In the next steps of the process, either the formation of the crystalline phase within the post-critical nucleus and subsequent crystal growth or the formation of a stable ACC may take place [16,106,107].

The concept of PILP, one of the non-classical mechanisms of calcium carbonate nucleation, was proposed in 2000 by Gower et al. [18]. They observed the formation of droplets of fluidic ACC precursor in the presence of negatively-charged polyelectrolytes during the calcium carbonate precipitation process with an in vitro model system. In the PILP pathway, nucleation is a multistep process where the polymer associates with calcium and bicarbonate ions to form an intermediate liquid phase prior to solid nucleation [109]. The liquid-like character of the early-stage amorphous precursor was evidenced by the coalescence of the droplets, which grow from tens of nanometres to a couple of microns, and by in situ AFM [18,110,111].

Based on the results of potentiometric titration and analytical ultracentrifugation (AUC) for undersaturated, saturated, and supersaturated solutions, Gebauer et al. proposed another concept that refers to prenucleation clusters (PNCs)—non-classical theory [1]. They observed that during the titration, the amount of free calcium ions was always less than the total calcium ions added, suggesting the formation of long-lived calcium carbonate clusters called PNCs. Once a critical point is reached, nucleation occurs, and the free calcium ions are consumed by the growing particles. Before nucleation, small cluster species with a hydrodynamic diameter of ~2 nm (corresponding to approximately 70 calcium and bicarbonate ions) were mostly detected in AUC, but the second-largest cluster species (hydrodynamic diameter of 4 to 6 nm) was also present, suggesting further nucleation via cluster aggregation. The proposed hypothesis was also confirmed by the fact that smaller clusters could not be detected in the post-nucleation phase [17,19]. The presence of PNCs has also been corroborated in solutions saturated with respect to calcite by cryo-TEM experiments; however, in contrast to the observations of Gebauer et al. [19], the obtained results showed that the prenucleation clusters persisted even after nucleation [112].

The amorphous precursor strategy refers to the approach by which organisms make use of the flexibility of ACC to control the kinetics of biomineral formation and the spatial distribution of the final calcium carbonate polymorphs. Despite the numerous examples of the transformation of synthetic and biogenic metastable ACC into a crystalline phase, the factors involved in the polymorph selection mechanisms as well as the effects of ACC precursors on the structural characteristics of the final products are still puzzling [113,114]. It was shown that stable hydrated ACC becomes dehydrated during transformation into the crystalline phase, which suggests that there might exist specific mechanisms involved in the stabilization, destabilization, and transformation of ACC involving some proteins and other ions [115]. It has been proposed that the formation of ACC in the precursor phase causes the lowering of interfacial free energy during the formation of crystalline phases [116].

Interestingly, recent studies increasingly use LLPS to explain the behaviour of calcium carbonate-containing solutions in the context of the mineralization mechanism [20]. Due to problems in determining the thermodynamics of the transient clusters that form during the nucleation process, molecular dynamics simulations were used to investigate the initial stability of the clusters with respect to the composition of the solution and formation pathway. Initially, high ion concentrations were used for the bioinformatics simulations to increase the frequency of ion association and to facilitate obtaining a cross-section of thermodynamic changes over time. Then, the clusters formed were transferred to a lower concentrated environment to demonstrate their stability under different conditions. It was found that the earliest formed clusters adopted chain, ring, and low-density branched structures. At high concentrations, growth occurred at the diffusion limit, with barriers opposing ion attachment with ambient thermal energy [117,118]. Low-density configurations were observed for small clusters. However, such configurations were quickly replaced by more condensed states with ion additives [104].

The dynamic nature of the clusters was quantitatively defined by the ion diffusivity components. The dependence of the diffusion coefficient of calcium ions within the clusters at different growth stages on the two solid phases of calcium carbonate, calcite and ACC, in several solvents has been described [119]. The ion diffusion properties were analysed in different solvents in the bulk ACC and calcite, indicating that the clusters are droplets of the dense, ion-rich hydrated calcium carbonate liquid phase. The ion diffusivity decreased with the increasing density of the liquid phase, but the rate of diffusion gradually decreased and approached a constant value characteristic of the depleted liquid phase. The lack of an energy barrier for cluster formation is characteristic of solutions that have passed their stability limit and undergo spontaneous phase separation via the spinoid pathway. The availability of the spinodal region, at low concentrations, is important for the mineralization process (Figure 3). Thermodynamically, this means that there is a line of liquid–liquid coexistence between the dense and depleted solution phases. Both liquids are in metastable equilibrium with respect to the calcium carbonate solid phases over a wide range of dissolution conditions (Figure 3). The solubility of all polymorphs is represented by a single solubility line (SL) that bounds the dashed unsaturated solution field. This representation highlights that all calcium carbonate solid phases (calcite, aragonite, vaterite, and presumably ACC) show the same general retrograde solubility behaviour. The nucleation of the solid phases progresses towards a high concentration on the dark blue line side of the liquid–liquid coexistence (L-L). The field between the binodal and spinodal lines bounded by the L-L line indicates the conditions under which nucleation of a dense liquid phase is possible. In the region bounded by the spinoid line, the solution is unstable, and liquid–liquid separation occurs spontaneously (Figure 3) [104].

The existence of a dense liquid phase makes it possible to apply both the classical model of ion crystallization and a model related to phase separation via clusters. Increasing the product of ion activity and the liquid–liquid coexistence line encounter makes homogeneous nucleation of a dense liquid phase possible. The formation of a dense liquid in a short time is more likely than direct crystallization because the excess of free energy at the solution–liquid interface is greatly reduced compared to the solution–crystal interface, thereby resulting in a lower thermodynamic barrier for liquid–liquid separation than for crystallization [104].

## 6. Liquid–Liquid Phase Separation in the Formation of Hard Tissue

Recently, research attention has been focused on assembly processes, including the formation of mineralized components of the body via LLPS. Faatz et al. were one of the first to show that spherical particles of ACC can be formed by LLPS [121]. Additionally, Wolf et al. demonstrated that ACC can grow in the absence of organic polymers [122]. In other studies, inorganic salts were shown to undergo LLPS at high temperatures [123,124]. In experiments in the presence of a poly-Asp additive, the data indicate that the polypeptide stabilizes a condensed phase of liquid-like droplets of calcium carbonate during PILP formation [109]. However, since liquid precursors can be detected in samples without any polymer additives, PILP may be considered a polymer-stabilized rather than polymer-induced state [109,122]. Therefore, an important question arises regarding the role of IDPs during the formation of biominerals: are IDPs inducers or modulators of the process? The available literature does not include many examples discussing their role in the formation of liquid-phase condensates and organic calcium carbonate components of the body. Recent studies on a nacre-like, aragonite-forming protein, Pif80, from *Pinctada funcata*, indicate that it has the ability to drive LLPS. Pif80 is a functional fragment of a Pif protein originating from proteolytic cleavage [125]. The protein has a high content of acidic residues; moreover, it exhibits a high degree of intrinsic disorder [126]. Bahn et al. showed that recombinant Pif80 (rPif80) underwent LLPS in the presence of calcium ions, thereby forming a dense protein-rich phase [127]. Pif80-containing liquid condensates occurred in solutions containing either chloride or bicarbonate ions as counter-ions, and the process was only mildly influenced by pH. These behaviours support the idea that electrostatic interaction is the major driving force for LLPS of Pif80. The mineralization experiments performed in conditions that allow for LLPS revealed the scenario in which Pif80 condensates and PILP-like calcium carbonate granules can coexist. Importantly, the PILP-like calcium carbonate granules contained an amorphous phase of the salt. Based on that, the authors suggested that the Pif80 condensates worked as stabilizing agents of PILP-like ACC inhibiting the growth of calcite. It is worth emphasizing that Pif80 was the first matrix protein for which the ability to undergo LLPS was shown [127].

As already mentioned, proteins capable of undergoing LLPS can be classified into four types: scaffolds (drivers), co-scaffolds (co-drivers), clients, and regulators [3,128]. This classification might reflect some functions of molecules (other than proteins) in biomineralization. Scaffolds are essential constituents of each condensate and are responsible for its integrity. In biomineralization, the role of scaffolds is assigned to collagen and chitin. On the other hand, co-scaffolds are components that need another co-scaffold to phase separate [129]. It is known that acidic IDPs in the organic matrix regulate the stability and polymorph selection of calcium carbonate at the molecular level [130]. They can induce nucleation, adsorb specifically onto some crystal faces, and/or intercalate in a controlled manner into the crystal lattice [84,131]. Some of them (e.g., rPif80) need an inorganic fraction (e.g., calcium ions) for LLPS [127]. Clients are dispensable components and reside in condensates only under certain conditions. This role might be assigned to carbonic anhydrases or calcium-binding proteins, for example [132]. The last type consists of molecules called regulators, which promote LLPS but are not located in the condensates (e.g., modifying enzymes) [128]. Many proteins involved in biomineralization are extensively post-translationally modified (e.g., phosphorylated, glycosylated, and proteolytically cleaved). For modifying enzymes, the regulatory role should be assigned [133,134].

## 7. Can LLPS Impact the Formation of Otoliths?

In our laboratory, we have been studying the molecular properties of proteins involved in the mineralization of fish otoliths and human otoconia for years. Similar to Pif80, proteins found in fish otoliths are negatively-charged polyampholytes (Figure 4) that can bind calcium ions. At present, there are no data in the literature indicating the ability of otolithic proteins to drive LLPS. Therefore, we performed in silico analyses to test the probability that LLPS is induced by OMM-64 protein from *Oncorhynchus mykiss;* Stm from *Danio rerio;* the Stm orthologue, Stm-l, from *Oryzias latipes;* and otolin-1 from *Danio rerio* (Figure 5).

At present, there are several computational tools for predicting the LLPS propensity of a given protein [136]. Each tool has a different approach and takes into consideration various characteristics, thereby enabling interactions that facilitate the formation of liquid condensates. The integrity of liquid condensates is maintained by weak and transient interactions between complementary binding partner regions. The possible interaction modes include charge–charge, cation-π, dipole–dipole, or stacking π-π matches of chemical groups [5]. For the in silico analysis of otolithic proteins, two predictors, FuzzDrop [137,138,139] and PScore [140], were chosen. The FuzzDrop algorithm is based on a model in which interactions within condensates are maintained by multivalent interactions between disordered regions, and a high score is given to regions that are unlikely to become ordered upon binding. This determines the local sequence bias with respect to composition, hydrophobicity, and structural disorder by examining a large number of possible sequence contexts [139]. On the other hand, the PScore approach gives the prediction of LLPS propensity based on the calculated likelihood of IDRs forming long-range planar π–π contacts [140].

According to the FuzzDrop predictor, the analysed otolith proteins involved in the formation of calcium carbonate biominerals showed a very high probability of driving LLPS. The calculated propensity for LLPS (pLLPS) equals 0.9991 for OMM-64, 0.9953 for Stm, 0.9946 for Stm-l, and 0.9969 for otolin-1 on a 0–1 scale. Such high values of pLLPS indicate that they may spontaneously undergo LLPS. Moreover, they may function as condensate drivers (pLLPS ≥ 0.6) [139]. The residue-based analysis revealed that except for otolin-1, almost all of the studied protein sequences had a high probability for LLPS. OMM-64, Stm, and Stm-l may be considered disordered polyelectrolytes [80,81,141], whereas collagen-like otolin-1 contains two externally-ordered domains. The central fragment of the protein possesses collagenous structures [85], while on its C-termini, a calcium ion-dependent C1q domain is present [89]. The C1q domains show no probable ability to drive LLPS, in contrast to the central region. To our knowledge, at present, there are no reports on the ability of collagenous proteins to drive LLPS. All four analysed proteins also contain regions with a potential tendency for context-dependent interactions. Such regions are able to adopt various binding modes, depending on their binding partner [142]. These regions may also contribute to the formation of an interaction network between proteins determining otolith morphology and material properties. The analysed proteins also contain some aggregation hotspots (not shown). In particular, OMM-64 contains several regions with a tendency to aggregate. In otoliths, it accumulates in ring-like structures. Tohse et al. found OMM-64 in high-molecular-weight aggregates (HMWAs) of the otolith matrix [79]. It is possible that the selected regions may be involved in the interaction leading to aggregation via the liquid-to-solid transition.

Otolith proteins were also analysed by the PScore predictor. This program predicts the likelihood of IDRs driving LLPS based on the propensity for long-range planar pi–pi contacts. This type of interaction is typically linked to the interactions between aromatic rings. Analysed herein, proteins are depleted in aromatic residues, but the rationale for using this predictor was the fact that orbitals of bonded sp2-hybridized atoms are present in other chemical groups of proteins, including amide groups, carboxyl, and guanidinium [140]. Therefore, proteins containing small residues allowing an exposition of the protein backbone are quite likely to form these contacts. Examples exist in which LLPS-driving regions are those that contain repeats enriched with small residues such as Gly-Pro or Gly-Arg residues [5,140]. The otolithic proteins analysed herein are rich in Gly and Pro residues, but as presented in Figure 4, OMM-64, Stm, and otolin-1 contain only short fragments, which obtained positive results in the PScore analysis. The Stm-l protein has no such region.

To summarize, according to our in silico analysis, otolithic proteins may have the potential to drive LLPS. Considering what we have recently learned about the importance of spontaneous LLPS in biological systems, it is likely that otolithic IDPs also drive the formation of dense condensates. At present, however, the significance of that potential ability remains to be solved.

## 8. Conclusions

Biomineralization leads to the formation of stiff components of the body that function as structures where inorganic salts form crystals, which are incorporated into the complex organic mesh. The presence of an organic matrix in biominerals influences more than just its material properties. Organic compounds, among which IDPs play a major role, may induce nucleation, function as regulators of the gross volume of the biomineral, and determine the pattern of growth of the mineral phase. Although studied for years, the mechanisms by which organic components play a role in nucleation and growth in the formation of mineral bodily components remain under debate. Undoubtedly, a better understanding of this process holds promise for a variety of fields, including drug and cell-therapy engineering, cancer/tumor target engineering, bone tissue engineering, and other advanced biomedical engineering [143]. Organic compounds that could influence the shape, size, and properties of biominerals could be used to induce the formation of biominerals with improved and strictly desired properties. The presence of organic molecules can also affect the incorporation of contaminating metal by substituting calcium ions in calcite. The application of this approach is, for example, promising for the remediation of toxic or radioactive metals in environments where calcite is stable over the long term [144]. Moreover, since calcium carbonate is abundant in the oceans, as many organisms use it to produce protective shell structures or skeletal elements, a better understanding of biomineralization pathways may be important for environmental and climate changing studies [145].

LLPS seems to be a widespread mechanism for the supramolecular organization of molecules. It often facilitates the assembly of proteins, both intra- and extracellularly. It is a thermodynamically driven process that guarantees the harmony of intracellular processes and likely extracellular processes as well, including the formation of mineralized components of the body. Notably, LLPS has only recently been appreciated in biomineralization studies. At present, it appears that only the tip of the iceberg has been discovered in that regard, and more fascinating discoveries will come.

## Figures and Tables

**Figure 1 biomolecules-12-01266-f001:**
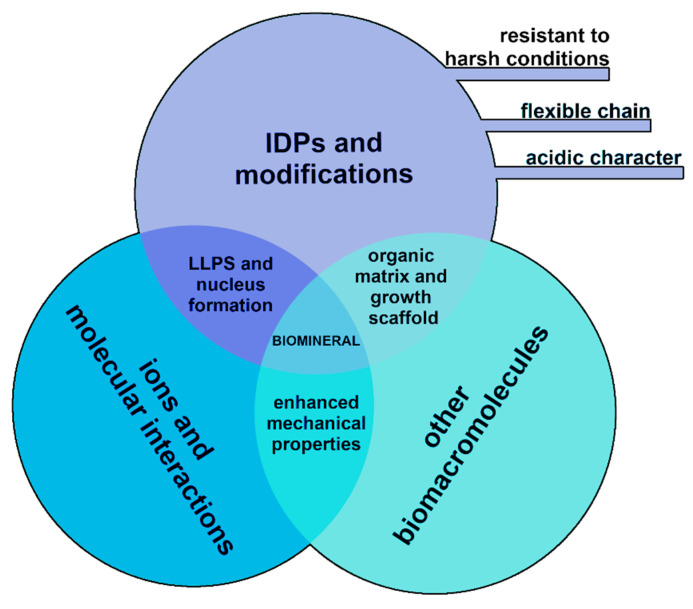
A conceptual diagram of the biomineral formation constituents and the relationships between them.

**Figure 2 biomolecules-12-01266-f002:**
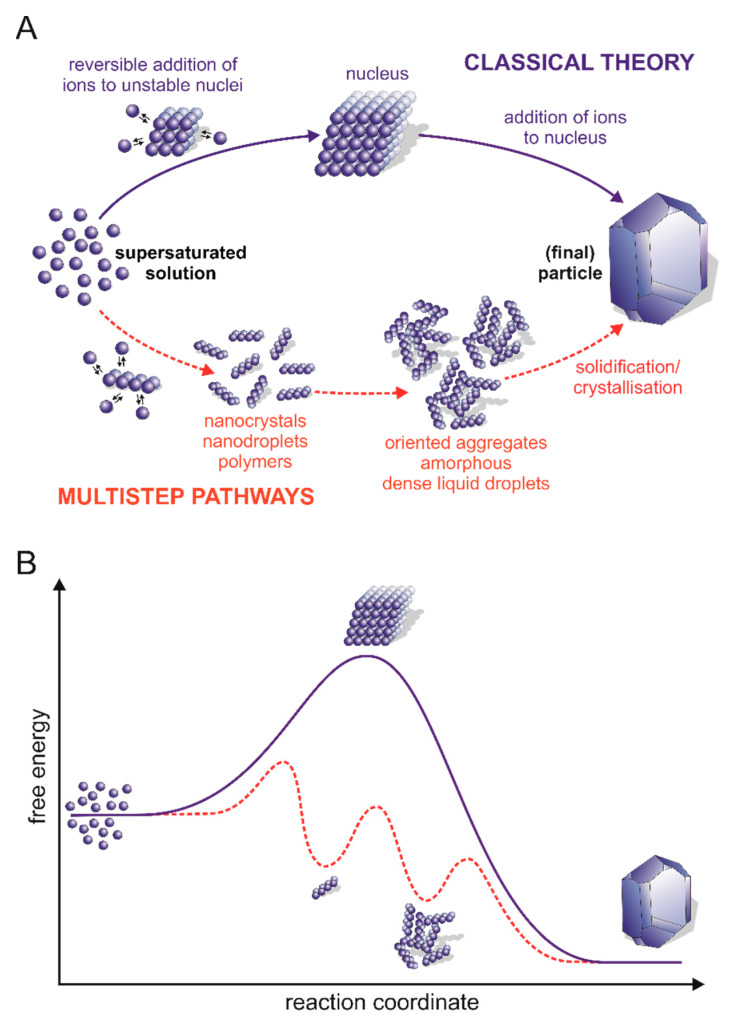
Classical and multistep nucleation pathways. (**A**) Schematic presentation of events occurring during the formation of crystals from the bulk liquid to crystalline pathways according to classical theory (blue) and multistep pathways (red). (**B**) Comparison of free energy along the pathway of crystal nucleation following the classical nucleation mechanism (blue) and multistep pathways (red) [20,108].

**Figure 3 biomolecules-12-01266-f003:**
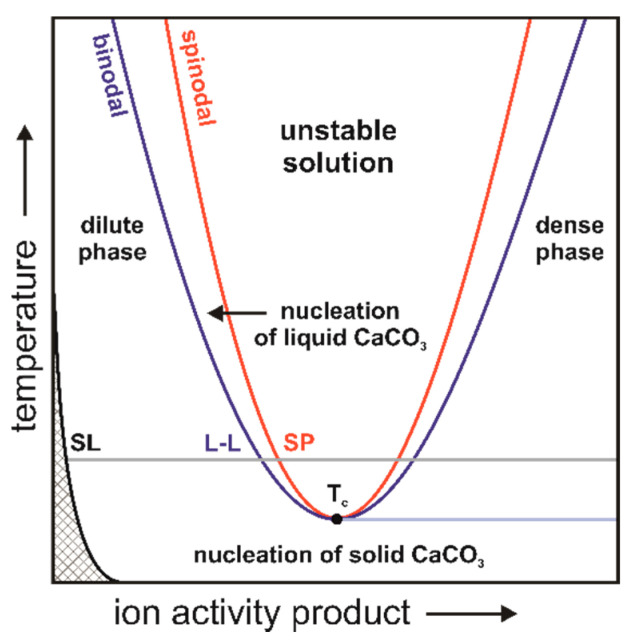
Schematic representation of the calcium carbonate phase relationships. The grey line represents an example solubility line (SL) of different polymorphs at constant temperature, the grey, checkered field corresponds to the unsaturated solution field, the L-L line represents the binodal (blue) and SP spinodal (red) lines, and T_c_ is the critical temperature point [120].

**Figure 4 biomolecules-12-01266-f004:**
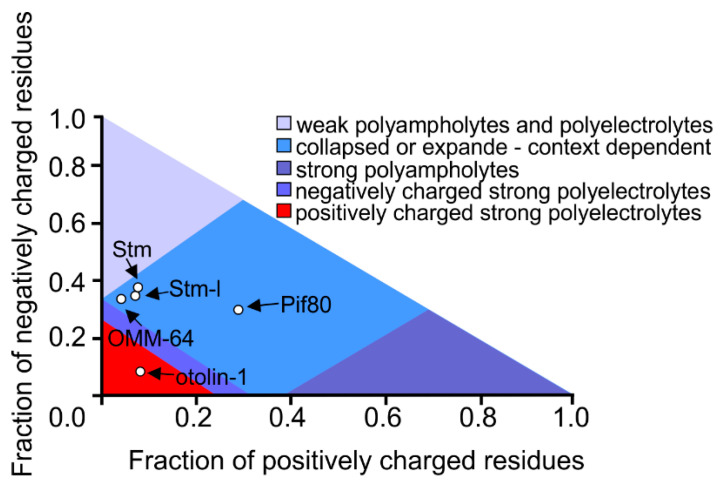
Phase diagram of proteins involved in the formation of fish otoliths. For the CIDER analysis [135], the following sequences were used: OMM-64 A0A060XQP6, Stm A2VD23, Stm-l A0A3B3H599, otolin-1 A5PN28, and Pif80 C7G0B5 (544-1007 aa residues [121]). [Accessed on 11 July 2022].

**Figure 5 biomolecules-12-01266-f005:**
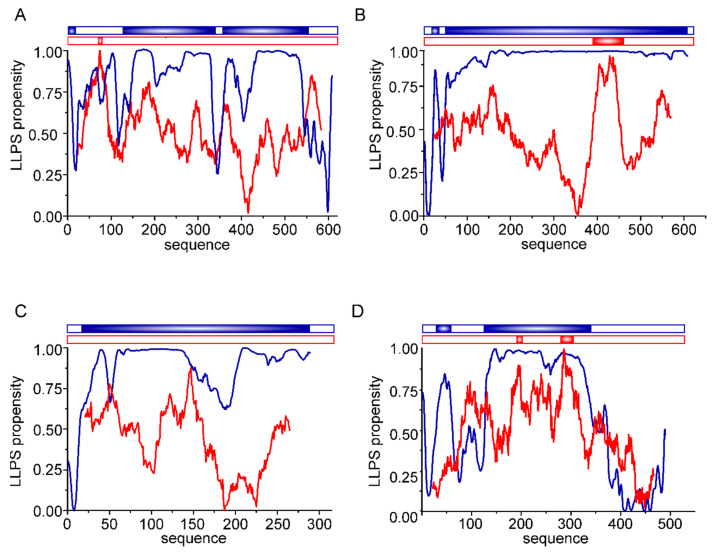
In silico analysis of otolithic proteins. The graph demonstrates the normalized results of in silico analysis of representative otolithic proteins: (**A**) OMM-64, (**B**) Stm (**C**), Stm-l, and (**D**) otolin-1 performed using FuzDrop (blue) and PScore (red). The protein regions that obtained positive scores are indicated as a scheme at the top of each panel. [Accessed on 12 May 2022].

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
