# Peer review of "The Role of Intrinsically Disordered Proteins in Liquid–Liquid Phase Separation during Calcium Carbonate Biomineralization"

_biomolecules, 2022, doi:10.3390/biom12091266_

Round 1

Reviewer 1 Report

Another excellent publication produced by this group.

This review, by Tarczewska et al. is both engaging and comprehensive. The roles played by IDP in biomineralization is a topic of keen interest, for both biochemists and fish ecologists. As my own area of research is chiefly focussed on otolith biomineralization pathways, I found the material presented here both well organised and thorough. I have a few minor comments to make, that can be addressed prior to publication:

Is it important to cite Davis et al. 1995 when referring to Otolin-1? I ask because you've cited all the other big players (Murayama, Tohse, Sollner etc.)

Line 159: It's probably worth citing Tagliabracci et al. (2012) here, as they did quite a lot of extensive work on characterisation of Extracellular Ser-Thr FAM20C Kinase and its role in phosphorylation of biomineralization proteins.

Line 209: Can you make more of phosphorylation and glycosylation's role on IDP function? I know your group has published extensively in this area. (Also, I see that you refer somewhat to it later (Lines 263-268), but I think there's more you can say.)

As a question, isn't it thought that phosphorylation of IDP causes negative charge sites on the surface of the protein, that can interact effectively with Ca2+, aiding alignment and polymorph selection, and that insufficient phosphorylation (either not at all, or a limited amount), can lead to different polymorphs being deposited in growing otoliths (i.e. vaterite or calcite instead of aragonite in sagittae)?

Farmed salmonid sagittae tend to be vateritic. Here, the animals are grown in low-carbohydrate diet conditions, which would lead to a lower ATP pool - and thus reduction of phosphorylation of key proteins (e.g. Stm-like).

Other than these minor remarks, I found this review to be exceedingly well written. The graphs and figures were logical, and the narrative flow was succinct.

Reviewer 2 Report

In this review, the authors described the role of IDPs in the biomineralization process, offering a new point of view of this phenomenon as a LLPS supramolecular organization. The authors well organized the review. After a short but comprehensive introduction about the three main subjects (IDPs, LLPS, biominerals), the authors first presented the role of proteins and bivalent cations in LLPS, then, the implication of IDPs in the process of biomineralization and finally an explanation of the two main hypotheses concerning the mechanism of biomineralization. This organization allows the readers to approach to the topic in a logical way and helps the comprehension. The new point of view given by the authors is understandable and clearly presented.  Only few suggestions to improve the review and make it more interested in a larger audience.

1.       In paragraph 5, when the authors explained the nucleation process hypotheses, I suggest making a comparison between the hypothesized mechanisms of nucleation with the widely known processes of protein aggregation, especially the ones involving IDPs and LLPS. Recently, the LLPS theory is growing as a new possible explanation of amyloid formation, whose process has a lot of similarities with the biomineralization presented by the authors.

2.       In paragraph 5, I recommended to better explain the physical-chemical approaches that have been used to show the possible mechanism. In the first part, a general overview is offered but it is not clear how the findings have been obtained to support the hypotheses.

3.       Could the authors try to better explain the importance to improve the knowledge of this phenomenon regarding possible biomedical or biomaterial applications? From the review it is not very easy to catch the impact of this phenomenon.

4.       The last part (Can LLPS impact the formation of otoliths?) is, for my personal point of view, not necessary. It’s a summary of the last results already published by the authors. I think this is another example that can be easily included in the sixth paragraph, in support of the LLPS hypothesis for the hard tissue formation.
